# Neuroimmune Response Mediated by Cytokines in Natural Scrapie after Chronic Dexamethasone Treatment

**DOI:** 10.3390/biom11020204

**Published:** 2021-02-02

**Authors:** Isabel M. Guijarro, Moisés Garcés, Pol Andrés-Benito, Belén Marín, Alicia Otero, Tomás Barrio, Margarita Carmona, Isidro Ferrer, Juan J. Badiola, Marta Monzón

**Affiliations:** 1Research Centre for Encephalopathies and Transmissible Emerging Diseases, Institute for Health Research Aragón (IIS), University of Zaragoza, C/Miguel Servet 155, 50013 Zaragoza, Spain; isabelmariagt91@gmail.com (I.M.G.); moisesgarces1@gmail.com (M.G.); belenm@unizar.es (B.M.); aliotgar@hotmail.com (A.O.); tbarrio_sc@hotmail.com (T.B.); badiola@unizar.es (J.J.B.); 2Departamento de Patologíay Terapéutica Experimental, Universidad de Barcelona, 08907 Barcelona, Spain; pol.andres.benito@gmail.com (P.A.-B.); mcarmona@idibell.cat (M.C.); 8082ifa@gmail.com (I.F.)

**Keywords:** scrapie, cytokines, dexamethasone, neuroinflammation, prion diseases

## Abstract

The actual role of prion protein-induced glial activation and subsequent cytokine secretion during prion diseases is still incompletely understood. The overall aim of this study is to assess the effect of an anti-inflammatory treatment with dexamethasone on different cytokines released by neuroglial cells that are potentially related to neuroinflammation in natural scrapie. This study emphasizes the complex interactions existent among several pleiotropic neuromodulator peptides and provides a global approach to clarify neuroinflammatory processes in prion diseases. Additionally, an impairment of communication between microglial and astroglial populations mediated by cytokines, mainly IL-1, is suggested. The main novelty of this study is that it is the first one assessing in situ neuroinflammatory activity in relation to chronic anti-inflammatory therapy, gaining relevance because it is based on a natural model. The cytokine profile data would suggest the activation of some neurotoxicity-associated route. Consequently, targeting such a pathway might be a new approach to modify the damaging effects of neuroinflammation.

## 1. Introduction

Scrapie is considered the prototype of prion diseases, which are a group of neurodegenerative disorders caused by the conversion of a cellular protein into a pathological isoform called prion.

Neuroinflammation is currently a widely accepted concept in neurodegeneration, particularly in prion diseases [1,2,3,4]. The neuroinflammatory process is defined as the prolonged activation of neuroglial cells with the corresponding production of inflammatory cytokines [5]. Consequently, there is a particular interest in investigating the roles of the innate and adaptive immune systems in several neurodegenerative disorders, with neuroglia as a key element in the neuropathological process [6,7,8].

A relevant number of studies have proposed a crucial role for cytokines as neuroinflammatory mediators in the cellular communication in these prion diseases [3,9,10,11,12,13,14]. The detection of these cytokines was described coinciding with the onset of clinical signs in both a murine model [9] and Creutzfeldt–Jakob disease (CJD) [15]. Moreover, a recent study described the presence of several genes implicated in inflammation that are upregulated in early phases of prion infection [16]. Nevertheless, although an altered profile of inflammatory intermediaries has been evidenced in some experimental murine models [9,14,15,17,18,19,20,21], scarce studies have focused on in situ tissue expression of these proteins [10,22], and none of them on a natural model.

Previous studies developed in scrapie-affected animals have led to conclusions about the glial role in the neurodegenerative progress that was extrapolated not only to other prion but also other neurodegenerative disorders [23,24,25]. More recently, this same in vivo model has been used to assess the changes of activation of glial cells associated with anti-inflammatory therapy [26]. This study constituted a powerful approach to the involvement of immune response in this neurodegenerative disease, confirming the occurrence of neuroinflammation in neurodegeneration. Specifically, a potential failure of astrocytes and a stimulation of phagocytosis of prion protein deposits by microglia were evidenced after dexamethasone (DEX) treatment. To examine the interglial communication mediated by cytokines in depth constitutes a main tool for advancement of the knowledge of how these mediators are really involved in neuroinflammatory mechanisms contributing to neurodegeneration [27,28,29,30]. It is indispensable to study the possible alteration of glial crosstalk that might enhance instead of prevent neuronal damage. Thus, it could be a crucial target for therapeutic approaches in prion diseases [31].

Overall, the actual role of prion protein-induced glial activation and subsequent cytokine expression during prion diseases is still incompletely understood. Consequently, to investigate the possible alterations of in situ cytokine expression in brain samples from animals naturally affected by scrapie and DEX treatment would be really helpful to determine whether these proteins could be significant factors in the progress of neurodegeneration in this group of diseases. Thus, the specific aim of this study is to assess the effect of the anti-inflammatory treatment on different cytokines which could be potentially related to neuroinflammation. Both immunohistochemical and expression patterns of different pro- and anti-inflammatory cytokines in several brain regions from treated and non-treated scrapie-affected sheep are compared in this study as a first step towards the ultimate goal that is to determine whether these proteins represent relevant targets in the immunopathogenesis of neurodegeneration.

## 2. Material and Methods

All the following experimental procedures were previously approved by the Ethical Committee of University of Zaragoza (Reference number: PI41/16, 03/10/2016). All efforts were made to minimize animal suffering during the experiments and to reduce the number of animals used.

All the analyses were performed on samples coming from animals included in a previously published study where, as cited above, the glial activation response in the progress of natural scrapie after chronic DEX treatment had been assessed [26]. All experimental details were provided in this previous manuscript, but briefly, a total of 25 sheep (age ranging from 4 to 10 years and all except for one of them with heterozygous alanine-arginine- glutamine and alanine-arginine-histidine, ARQ/ARH, presenting homozygous alanine-arginine- glutamine, ARQ/ARQ genotype) were included in this study: 10 healthy control (of which 4 treated and 6 non-treated) and 15 clinical scrapie Rasa Aragonesa ewes (10 treated plus 5 non-treated). Healthy controls were considered essential in order to specifically observe the effect of treatment (daily intramuscular, IM 0.04 mg/kg dose until euthanasia by endpoint criteria, 16 months the longest) in normal conditions in ovine species. After euthanasia with intravenous pentobarbital injection, necropsy of each sheep was performed and 80 samples were subsequently collected and distributed for different studies. One hemi-section from each sample was fixed by immersion in 4% paraformaldehyde for immunohistochemical studies and the other hemi-section was frozen at −80 °C for molecular studies (RT-qPCR).

### 2.1. Immunohistochemical Techniques

Immunohistochemistry (IHC) was carried out in order to assess in situ neuroinflammatory profile associated with DEX treatment in all sheep. It was compared with non-treated sheep group in four encephalic areas (frontal cortex: Fc, cerebellum: Cb, obex: O and medulla oblongata: MO). 

Prior 4 µm sectioning, paraffin-embedding of fixed samples was developed. After specific pre-treatments for antigen retrieval, specific immunohistochemical protocols by using specific primary antibodies against those cytokines or their receptors mainly studied in literature related (to our knowledge, IL-1α, IL-1R, IL-2R, IL-6, IL-10R, TNFR and IFNγR) were applied. EnVision system (DAKO, Glostrup, Denmark) and diaminobenzidine (DAB; DAKO, Glostrup, Denmark) were used as the visualization system and chromogen, respectively. Hematoxylin counterstaining and mounting in DPX was finally performed on all sections.

All slides were analysed by two independent observers scoring the intensity of immunostaining from 0 (absence) to 4 (maximum presence) by counting positive cells [32] in 5 microscopic fields in each brain region examined. Moreover, provided that cerebellum has been proposed as a pseudo reference region to detect neuroinflammation [33], close attention was paid to different profiles of cytokine distribution in this brain region. The focus was on Purkinje cells based on previous results referring this neuronal type as the most damaged while they are the most protected neurons in this area [23,26,34]. Table 1 summarizes all the primary antibodies and the protocols applied as following described.

#### 2.1.1. IL-1, IL-1R, IL-6 and IFNγR Detection

A pre-treatment consisting of hydrated heating at 121 °C in citrate buffer 10% for 20 min preceded the endogenous peroxidase blocking (DAKO, Glostrup, Denmark) for 5 min and incubation overnight 4 °C with different primary antibodies: polyclonal IL-1α (1:100; ThermoFisher Scientific, Waltham, MA, USA), polyclonal IL-1RN (1:100, Sigma, St. Louis, MO, USA), monoclonal 8H12 (1:40; ThermoFisher Scientific, Waltham, MA, USA) or polyclonal IFNGR1 (1:200, ThermoFisher Scientific, Waltham, MA, USA).

#### 2.1.2. IL-2R, IL-10R and TNFR Detection

A pre-treatment consisting of hydrated heating at 96 °C in citrate buffer 10% for 20 min preceded the endogenous peroxidase blocking (DAKO, Glostrup, Denmark) for 5 min and incubation with different primary monoclonal antibodies: IL-2R.1 (1:1000, overnight 4 °C; ThermoFisher Scientific, Waltham, MA, USA), OTI1D10 (1:250, overnight 4 °C; ThermoFisher Scientific, Waltham, MA, USA) or Ber-H2 (ready to use, 30 min RT; DAKO, Glostrup, Denmark).

### 2.2. RT-qPCR

Cerebellum and frontal cortex frozen tissues from treated and non-treated scrapie animals were included in the following comparative molecular analysis for some inflammatory markers.

#### 2.2.1. RNA Purification

The purification of RNA was performed following the instructions of the supplier (RNeasy Lipid Tissue Mini kit, Qiagen, GmbH, Hilden, Germany). RNA integrity and 28S/18S ratios were determined with the Agilent Bioanalyzer (Agilent Technologies Inc, Santa Clara, CA, USA). Samples were treated with DNase digestion, and RNA concentration was evaluated using a NanoDrop Spectrophotometer (ThermoFisher Scientific, Waltham, MA, USA). 

RNA samples with OD 260/280 ratios close to 5.0 were selected for reverse transcription. Finally, a total of 5 treated and 4 non-treated clinical sheep were included in this molecular analysis. 

#### 2.2.2. Retrotranscription

Retrotranscription of RNA into cDNA was performed according to the manufacturer’s instructions (High-Capacity CDNA Reverse Transcription Kit, Applied Bio systems, Foster City, CA, USA).

#### 2.2.3. RT-qPCR

Gene expression of IL-1, IL-6, IL-10Ra, IL-10Rb and IFNγ in both clinical non-treated and treated sheep was assessed. The parameters of the reactions were 50 °C for 2 min, 95 °C for 10 min, and 40 cycles of 95 °C for 15 sec and 60 °C for 1 min. 

Data were assessed using the ΔΔCt method, using Hypoxanthine Phosphoribosyl transferase 1 (HRPT-1) and β-glucuronidase (GUS-β) as reference genes. Table 2 shows TaqMan probes used for these molecular studies.

### 2.3. Statistical Analysis 

For IHC results, the normality of distribution was first assessed by Kolmogorov-Smirnov test. The non-parametric Mann–Whitney *U* test was used to assess quantitative differences between non-treated and DEX treated groups. 

Data of RT-qPCR were evaluated by Student’s *t* test after assessing normality also by Kolmogorov–Smirnov test.

SPSS software (SPSS Statistics for Windows, Version 17.0, Chicago, IL, USA) was used for these analyses and significance in all cases was taken at * *p* < 0.05. All graphs were performed with GraphPad Prism 6.0 (San Diego, CA, USA). Data presented in figures are expressed as means and the standard error of the mean (mean +/− SEM).

## 3. Results

### 3.1. Immunohistochemistry

#### 3.1.1. IL-1

DEX-treated controls always displayed higher intensity for IL-1 staining compared to the untreated samples. However, clinically treated animals showed lower differences compared to their respective untreated group and were even reversed (Figure 1A). 

The Mann–Whitney *U* test revealed significant effects of DEX in treated controls compared to untreated samples, showing an increase of IL-1 immunostaining in MO (* *p* = 0.024). Meanwhile, no changes were detected in clinically treated scrapie sheep depending on the treatment (Figure 1B). 

In Cb, immunostaining for IL-1 was widespread in all layers (Figure 2A). 

#### 3.1.2. IL-1R

In general, immunostaining was lower in treated animals compared to non-treated ones, except for controls in Fc, where the patterns were exactly the opposite (higher). These differences evidently increased in O and Cb at clinical stage (Figure 3A). 

Despite the fact that there were no statistically significant changes observed for this marker, a trend to a reduction in immunostaining in Cb from animals in the clinical stage was detected (# *p* = 0.082) (Figure 3B).

IL-1R immunoreactivity in Cb was nearly exclusively located in cells surrounding Purkinje cells, suggesting morphology consistent with specific astrocytes (Figure 2B). 

#### 3.1.3. IL-2R

It is only relevant to point out that Fc showed an exacerbated increase of reactivity against this cytokine, regardless of treatment, and disease, in comparison with the other brain areas. 

No statistically significant differences were found regarding treatment in both the control and clinical groups (Figure 4A). 

In Cb, IL-2R was limited to the cytoplasm of Purkinje cells and cells with an astrocytic appearance (Figure 2C). 

#### 3.1.4. IL-6

Immunostaining for IL-6 in DEX-treated controls was higher in all brain areas compared to untreated controls, except in O, where the labeling intensity was lower. This reduction in O was more evident at clinical stages of disease (Figure 5A). 

The increase of immunostaining intensity reached significance in Fc of the control group when animals were DEX-treated (* *p* = 0.040). Meanwhile, the immunoreactivity for this cytokine showed a decreasing trend in control animals (# *p* = 0.071) that converted into significant in O in the clinical group (* *p* = 0.041) (Figure 5B). 

This neuromodulator peptide was mainly present in Purkinje cells as intracytoplasmic staining (Figure 2D). Moreover, the cytokine detected was present in specific cells resembling glial cells in both granular and molecular layers.

#### 3.1.5. IL-10R

Immunostaining for this marker was the highest in all assessed samples in both the control and clinical groups, regardless of treatment, but especially in Fc, where the highest score was reached regardless of treatment or disease.

Along the same lines, although not significant, a trend toward a decrease (# *p* = 0.074) in Cb of clinical sheep after treatment was observed (Figure 6A). In general, statistically, immunostaining for IL-10R did not reveal any changes between groups after treatment (*p* > 0.05). Nevertheless, a very relevant decrease in O intensity in treated clinical animals was evidenced to reach a lower intensity than untreated, while the staining intensity was higher in controls (Figure 6B). 

This marker stained Purkinje neurons consistent with a spot intracytoplasmic pattern. Additionally, the staining showed a high intensity in granular cells and some others with astrocytic morphology (Figure 2E).

#### 3.1.6. TNFR

In general, immunostaining for TNFR was less intense in all brain areas compared to the rest of the cytokines assessed here. 

The pattern of TNFR immunoreactivity was similar in the control and clinical animals, with no significance after treatment in any case (*p* > 0.05) (Figure 4B). 

Morphologically, Purkinje cells in Cb expressed this receptor at a much lower extent than the rest of the cytokines assessed here (Figure 2F). 

#### 3.1.7. IFNγR

Whereas in DEX-treated controls, the immunostaining was higher in O and MO, this difference was reversed in the DEX-treated clinical group. In contrast, immunostaining intensity increased in Cb from treated controls to exceed that of untreated animals when they were affected by clinical disease.

Nevertheless, no significant differences in intensity levels were found between treated and non-treated groups in any encephalic area for this marker (*p* > 0.05) (Figure 4C) 

In Cb, its immunostaining pattern was mainly located in the cytoplasm and dendritic spines of Purkinje cells (Figure 2G). 

### 3.2. RT-qPCR 

Significantly, DEX treatment resulted in more than a three-fold increase in IFNγ mRNA levels in Fc of clinically treated sheep (* *p* = 0.036) (Figure 7A). 

Regarding Cb, no statistically significant changes were found in the mRNA expression of any cytokine analyzed here, although all of them were higher in clinically treated sheep (Figure 7B). Table 3 represents a summary recapitulative table of the results provided by application of the statistical analysis. 

## 4. Discussion

A great number of suggestions have led to hypotheses about the existence of a mixed inflammatory profile in prion diseases contributing to immune response [35,36,37]; on the one hand, some studies described a limited pro-inflammatory response, with no increases in IL-1 and TNF [18,22,38,39]; on the other hand, some others described increases of some factors, these same cytokines among them [9,10,15]. Actually, the presence of an impaired response of cytokines in mice orally infected with scrapie, imitating the natural route of entry, has been described [32]. However, these previous investigations in murine models have not been able to reach a consensus regarding the inflammatory profile in prion diseases because expression levels vary depending on the strain of mice and different combinations of scrapie inoculum [14,38], as well as different disease stages and detection techniques [37]. Moreover, direct detection of cytokines in the brain by IHC has been achieved only by some authors, probably because of the challenge associated with their detection as labile soluble proteins present at low concentrations [40]. A novel in situ cytokine profile is described here, slightly different from what has been previously described, possibly because the natural model was used here. It is crucial to highlight the relevance of results from natural models providing a fairer reflection of reality, since they are more reliable than those from experimental models mainly based on immunological discrepancies between sheep and rodent models even independently of prion diseases.

The findings presented in our previous study supported an impaired astroglial response at the clinical stage of scrapie, suggesting astroglial paralysis [26], as was recently described in late stages of Alzheimer’s disease (AD) [41]. Our next goal consisted of providing specific information about the mechanisms underlying cytokine release from glial cells. For that purpose, the effects of long-term DEX treatment in sheep with natural scrapie and in healthy controls were examined here to determine whether this treatment would change the levels of neuroinflammatory markers in this neurodegenerative disease. To clarify the relationship between neurodegeneration and neuroinflammation is the ultimate goal.

As it has been reported that neuroinflammation in prion diseases is region-dependent [42], this study assessed four different brain areas. Moreover, no studies about in situ variations in cytokine levels after glucocorticoid (GC) treatment in prion diseases have been previously published, despite the fact that it is well known that chronic administration of GC inhibits both innate and adaptive immune systems, reducing a great variety of pro-inflammatory cytokines [43], such as IL-1, TNF [44], IL-6 and IFNγ [45,46]. In most cases, chronic exposition to GC suppresses mediators; however, on some occasions, GC can potentiate such immunity [47], especially in the central nervous system, which can exacerbate neuronal death [48]. Thus, in some cases of murine AD, neurodegeneration and exacerbation of pro-inflammatory mediators were observed after chronic exposition to this anti-inflammatory drug [49]. Consequently, the doses and duration of treatment are of huge importance for the immune system response. Additionally, acquired resistance with absence of response to treatment has also been described in rheumatoid arthritis [50].

Here, we show that the low dose of DEX chronically administered to healthy animals significantly increased the secretion of IL-1, specifically in one of the brain areas examined (MO). The early over-expression of IL-1 has been suggested as a candidate to contribute to the development of murine [1,14,19] and ovine scrapie [3], as well as AD [51,52,53,54]. This increase has been also described in clinical and terminal stages in another scrapie murine model [22]. Furthermore, it is a well-known fact that expression of IL-1 induces oxide nitric synthase (iNOS) [55], and this production is involved in damage and neuronal death [56]. Enhanced neurodegeneration was observed when anti-inflammatory treatment was provided before the development of the neurodegenerative lesion itself in rats [57]. Taking into account this enhancement, DEX could be speculated to present a possible neurotoxic effect in control sheep. As our previous study about glial activation response after DEX treatment showed an increment in glial fibrillary acidic protein (GFAP) levels in treated controls, mainly observed also in MO [26], higher IL-1 levels could be related with to astrocytic activation, provided that some authors have closely related IL-1 to astrogliosis [1,14]. Conversely, no variations in the level of this mediator were evidenced after treatment when animals were scrapie-affected. This could reinforce the hypothesis that an astroglial dysfunction exists in scrapie progress [26]. An impairment of communication might be suggested between microglia and astroglia in clinical scrapie because IL-1 is not increased in this group of animals, although microglial population is shown to be activated [26]. 

In this natural model, the expression of IL-1R was slightly reduced in all treated cases (both control and clinical), except in Fc of controls. As that deficiency of IL-1R was demonstrated to prolong incubation time [58], delay disease onset and expand survival period of affected mice [19], this could point to the possibility of a neuroprotective role for DEX in both control and clinical sheep. The presence of neurotoxic reactive astrocytes induced by IL-1 and TNF released by microglia has been described [59]. Furthermore, a recent study showed that microglia have the power to auto renew themselves and that IL-1R signaling is implicated in this proliferative process [60]. Despite further studies to go in depth into this issue would be necessary to be performed, it could be speculated that DEX prevents the self-renewal of microglia, which might be related to the impairment of communication with astroglia observed in our previous study [26]. Additionally, IL-1R immunoreactivity was located mainly surrounding Purkinje cells in the cerebellum and appeared morphologically consistent with that of astrocytes, in agreement with previous studies in murine models [10,11,14,15]. Indeed, IL-1R has been described to be expressed by microglia, astrocytes and neurons in rodents [61,62], but particularly by cerebellar Purkinje neurons [62]. Importantly, it appears that IL-1R in neurons mediates the fast changes in neuronal excitability induced by IL-1, while IL-1R in astrocytes mediates effects of IL-1 on neuronal survival [63]. Therefore, although future experimental studies for separately analyzing expression of levels of IL-1R of astrocyte or microglia are required, evidence of the complexity of the IL-1 ligand/receptor, whose balance depends on the phase of the inflammatory response, is shown here [64]. Given that only few activated IL-1R molecules per cell are sufficient for activation of the target cell [65], this is of particular relevance for their signaling system. Overall, it is strongly evidenced here that both IL-1 and IL-1R play significant roles in the progress of natural ovine scrapie. 

Regarding IL-2, it has been shown that this mediator can ameliorate amyloid pathology in mice with AD [66]. Nevertheless, neither this mediator nor its receptor have been analyzed in the prion diseases field before. Some trials revealed IL-2 anti-inflammatory effects [67]. However, in the present study, no differences were found regarding treatment in both the control and clinical groups, thus not attributing a relevant anti-inflammatory role of DEX regarding this cytokine. Fc has been shown to be the more reactive brain region against this mediator, and, similar to our study, it has been shown that immunoreactivity of IL-2 in the brains of both controls and AD patients was widespread [68]. Morphologically, in Cb, its receptor was generally found in the cytoplasm of Purkinje cells and astrocytes. Given that other authors have described that hippocampal neurons are rich in this receptor, which has been related to neuronal development [69,70], we could also suggest that IL-2 is involved in neuronal development of Purkinje cells in this encephalic area. Taken together, although DEX treatment has had no apparent effect in modulating the levels of this peptide, a role of IL-2 in prion diseases should not be discarded before developing further studies.

Whereas DEX caused a significant increase in IL-6 staining intensity in Fc of the control group, which is in complete agreement with the strong astrogliosis observed in treated controls in our previous study [26], the immunoreactivity for this cytokine was reduced in O of both the treated control and the clinical group. Of particular interest is the fact that over-expression of IL-6 has been shown to produce neurologic disease in mouse, activating astrocytes and microglia [71]. Moreover, Burwinkel et al., 2004 [1] suggested that IL-6 might be implicated in the terminal stage of scrapie since it activates astrocytes in vivo [72]. Consequently, blocking of IL-6 has been proposed as a possible treatment against chronic inflammatory diseases [73]. Taking all this into account and given the evidence presented herein, DEX might act as neuroprotective for both control and clinical animals in one brain area (O), while it could be neurotoxic for controls in another area (Fc). Needless to say, IL-6 is a very peculiar cytokine that, depending on the context, exhibits two different behaviors: in acute inflammation models, it is anti-inflammatory [74,75], while it is pro-inflammatory in chronic inflammation models [76,77]. In summary, IL-6 can act as both pro- and anti-inflammatory, mediating both neurotrophic and neurotoxic effects on neurons [78]. Thus, although one study in mice postulated that IL-6 gene expression deficiency did not have any effect in prion pathogenesis [79], we strongly support the idea that this neuromodulator peptide is relevant to prion disease pathogenesis [3,9,10,11,12,13,14,15], as has been described for prion-*like* diseases [80,81,82].

Immunostaining of the marker specific for IL-10R was the highest compared to the rest of cytokines, which is in accordance with the treatment applied, as this mediator has been described as a potent anti-inflammatory cytokine [83] that interferes with nuclear factor kB (NF-kB) activation and blocks the synthesis of other pro-inflammatory cytokines [84], thus receiving considerable attention in neuroinflammation research [85]. Actually, Thackray et al., 2004 [20] demonstrated an accelerated pathogenesis of prion diseases in the absence of IL-10, with extension of survival time in mice [58], which is attributed to a protective role of IL-10 in prion diseases. Nevertheless, as no changes between groups after treatment have been detected (nor by IHC nor RT-qPCR), and even subtle decreases in intensity of this mediator in some brain areas in the clinical stage after treatment were observed in the present study, the immunomodulatory and anti-inflammatory roles of this cytokine in natural ovine scrapie cannot be completely confirmed, as have been recently described in models of AD [86,87].

Any changes were not detected for TNFR in any group, regardless of treatment. This supports studies that did not attribute a relevant role to this cytokine in scrapie progress [79] and prion pathogenesis [88]. However, TNF has been described to be implicated in panencephalic CJD pathogenesis [17], and an increment in incubation period in TNF *knockout* animals has also been reported [89]. Indeed, TNF is a very peculiar mediator, as both elevated [90] and reduced levels [91] of this peptide and its receptor have been described in AD. Consequently, either neuroprotective [92] or neurotoxic roles [93] have been suggested by other authors [94]. The discrepancies among studies regarding the role of this cytokine might be due to the different experimental models used. Thus, it is worth mentioning that, to our knowledge, this is the first study based on a natural model assessing in situ neuroinflammatory activity in prion diseases in relation to chronic anti-inflammatory therapy. 

Regarding IFNγR intensity levels, a similar behavior was demonstrated in all groups. However, immunostaining was slightly higher in some encephalic areas (O and MO) from controls after treatment, while this difference was diminished and even reversed in the clinically treated stage. Taking into account that microglia stimulate their division and adult neurogenesis by releasing IFNγ, among other mediators [95,96], this finding could be reflecting an attempt for microglial division in some brain regions in the controls. Meanwhile, no effect was proved at the clinical stage of the disease by IHC. Regarding molecular expression, strikingly, we found a significant increase in the quantity of IFNγ mRNA by RT-qPCR in a specific area (Fc) of clinically treated sheep, and these results could be reflecting the microglial expansion observed in previous studies in natural scrapie [23,26]. However, no statistically significant changes were found in the mRNA expression levels of any cytokine in Cb, in agreement with the idea that neuroinflammation in prion diseases is region-dependent [42]. The discrepancies in results provided by molecular and IHC techniques are most likely due to the reduced number of samples available for RT-qPCR (9 samples in total, while 15 samples were used for IHC). In addition, as prolonged expression of IFNγ in central nervous system has been reported to lead to neuronal and glial cell damage [97], and activated astrocytes have been described to have high levels of IFNγR in Parkinson’s disease models, suggesting that this cytokine can mediate the toxic effects of neighboring neurons [98], DEX treatment could have had neurotoxic effects in both control and clinical animals. Further studies are necessary to clarify this discernible contradiction. 

Concerning observations focused on cell damage in Purkinje cells as the most damaged but most protected neurons in the cerebellum [23,24,26,99], different immunostaining patterns of cytokines or receptors have been demonstrated in this study. Certainly, other authors found that immunostaining levels for cytokine receptors were higher in Purkinje cells than in other cells in mouse cerebellum [100]. Such distributions in mice are similar to those described in our study in sheep, confirming that cytokine receptors are present at high extent in Purkinje neurons in ovine cerebellum. This distribution could mean that the respective ligand of each receptor can act directly on Purkinje cells to alter their physiological properties, which possibly means that cytokines might be involved in this cellular damage around Purkinje cells, reconfirming the previously proposed role of Purkinje neurons in neuroinflammatory processes [101].

Overall, the therapeutic approach that has been tested here was demonstrated to directly influence the neuroinflammatory response. The results provided by this study confirm a limited modulating effect of low-dose DEX on the release of immunomodulator peptides from glial cells in a natural sheep scrapie model and reaffirm the paradoxical immunomodulator functions that glucocorticoids as DEX exhibit [102,103].

## 5. Conclusions

This is the first study using a glial-directed strategy in a natural ovine model of scrapie to investigate the contribution of inflammatory responses mediated by cytokines. An impaired communication between microglia and astroglia, which is mediated by cytokines, with IL-1 as one of the main actors, is suggested here. This study also emphasizes the complex interaction among several pleiotropic neuromodulator peptides and provides a global approach to clarify the neuroinflammatory process in natural prion diseases.

The cytokine profiles described in this study suggest the activation of some neurotoxicity-associated pathways. Consequently, targeting such pathways might be a new approach to modify the damaging effects of neuroinflammation. Further studies to determine which specific cellular types are expressing these cytokines and how these proteins are involved in cell signaling leading to neurodegeneration are essential. 

## Figures and Tables

**Figure 1 biomolecules-11-00204-f001:**
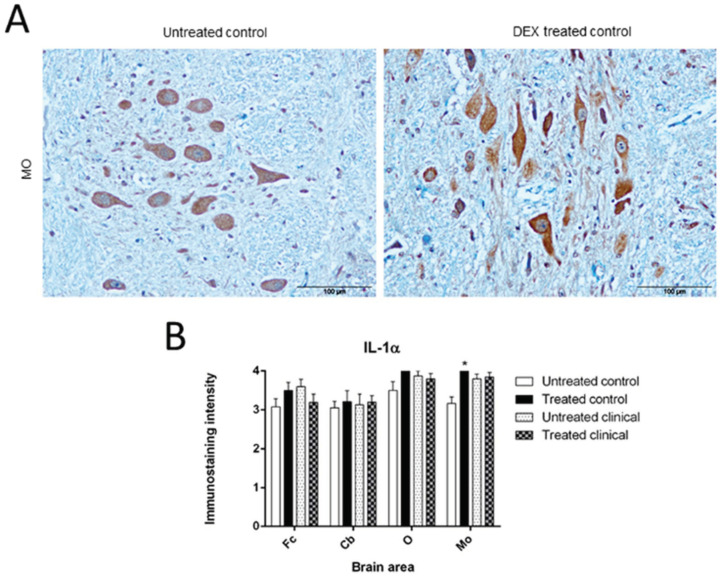
IL-1 immunostaining. (**A**) Note the evident higher intensity of immunostaining in medulla oblongata, MO from DEX treated control sheep compared to an untreated one. Scale bars: 100 µm. (**B**) A significant effect of DEX was observed in MO from treated controls (* *p* < 0.05). However, in clinical stage, no significant changes were detected despite of the treatment.

**Figure 2 biomolecules-11-00204-f002:**
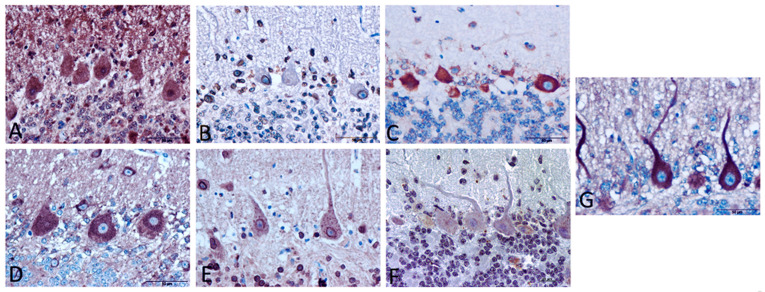
Morphological findings in cerebellum immunostained with different primary antibodies. (**A**) Immunostaining for IL-1 was widespread in all layers. (**B**) IL-1R immunoreactivity was located mainly surrounding Purkinje cells, suggesting a morphology consistent with astrocytes. (**C**) IL-2R was mainly found in cytoplasm of Purkinje cells and cells appearing astrocytes. (**D**) IL-6 immunostaining was mainly present in Purkinje cells as intracytoplasmic staining, as well as stained cells resembling glial cells in both granular and molecular layers. (**E**) IL-10R immunostaining appeared spot intracytoplasmic in Purkinje cells as well as granular cells and other cellular type with astrocytic morphology. (**F**) Purkinje cells expressed TNFR in a very lower extent than the rest of markers. (**G**) IFNγR immunostaining pattern was mainly localized in the cytoplasm and dendritic spines of Purkinje cells. Scale bars: 50 µm.

**Figure 3 biomolecules-11-00204-f003:**
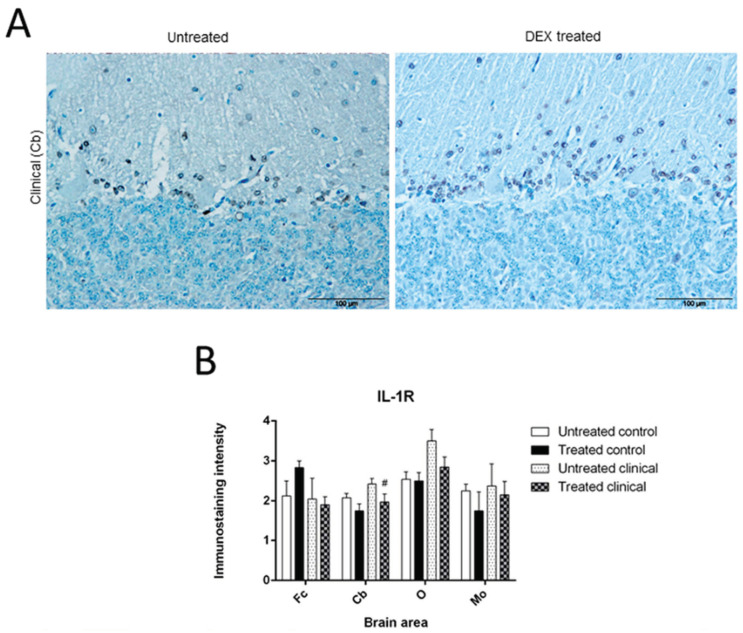
IL-1R immunostaining. (**A**) Differences observed after treatment in Cb at clinical stage are illustrated. Scale bars: 100 µm. (**B**) While a subtle reduction of IL-1R in treated control group was observed in the rest of areas examined, in Fc it increased after DEX treatment. At clinical stage, a decrease of immunostaining in O and a trend (# *p* = 0.082) to reduction in Cb were detected after treatment.

**Figure 4 biomolecules-11-00204-f004:**
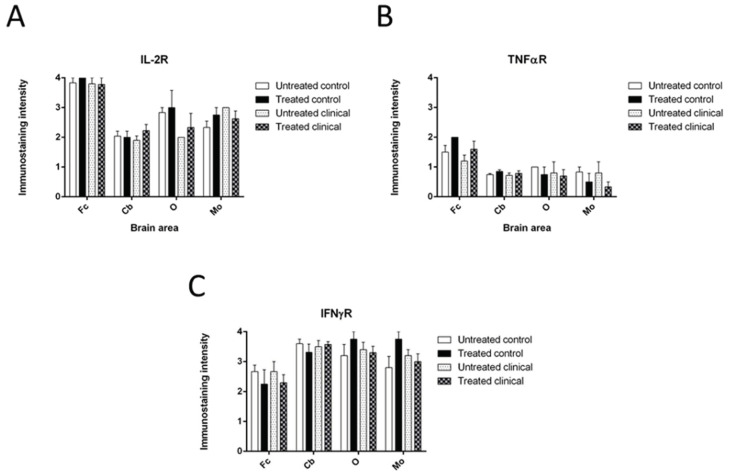
No significant differences were found either in control or in clinical groups for immunostaining intensity for (**A**) IL-2R (**B**) TNFR or (**C**) IFNγR. A higher intensity in obex, O and medulla oblongata, MO after treatment was evidenced in controls that reversed in DEX clinical animals.

**Figure 5 biomolecules-11-00204-f005:**
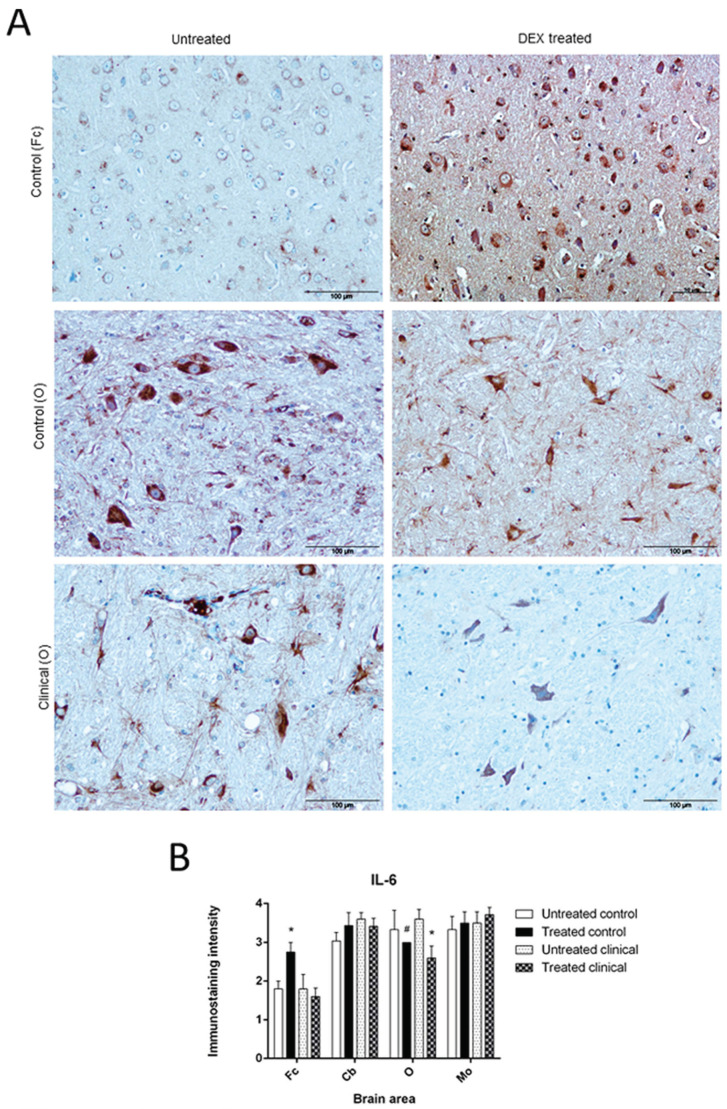
IL-6 immunostaining. (**A**) Micrographs represent an increase of IL-6 immunoreactivity in frontal cortex, Fc in treated controls while slightly reduced in both treated groups in obex, O. Scale bars: 100 µm. (**B**) DEX control group presented a significant increase in Fc (* *p* < 0.05) and a trend to a decrease in O (# *p* = 0.071). It was significantly decreased in this area when clinical group was treated (* *p* < 0.05).

**Figure 6 biomolecules-11-00204-f006:**
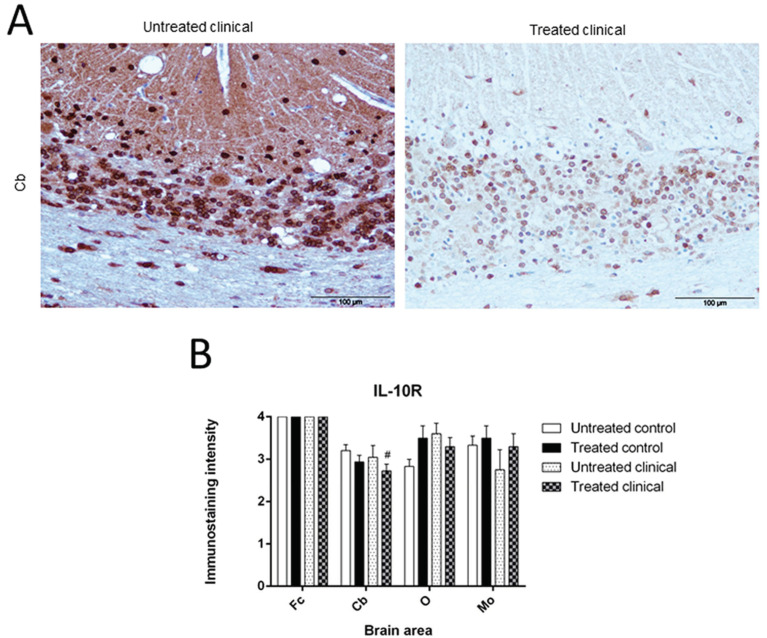
IL-10R immunostaining. (**A**) Decrease in intensity in Cb of treated clinical sheep is illustrated. Scale bars: 100 µm. (**B**) No significant changes were detected after treatment in control group, only a higher increase in O was outstanding. No significant changes were either observed in clinical stage after treatment, just a trend to lower intensity in Cb of treated clinical sheep (# *p* = 0.074).

**Figure 7 biomolecules-11-00204-f007:**
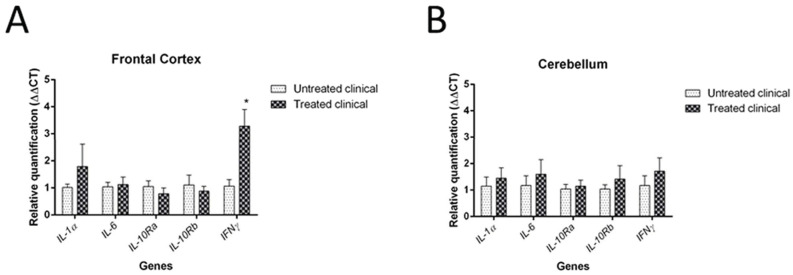
Results of gene expression by RT-qPCR in clinical sheep. (**A**) In Fc, an increase of IFNγ mRNA levels was observed (* *p* = 0.036). (**B**) In Cb, no significant changes were found in the mRNA expression of any cytokine analysed in this study, but all of them were slightly higher in clinical stage after DEX treatment.

**Table 1 biomolecules-11-00204-t001:** Antibodies used for immunohistochemical techniques and retrieval method applied for each one.

Antibody	Antigen	Type	Dilution	Retrieval Method	Source
IL-1 alpha	IL-1	Polyclonal	1:100	Autoclave 121 °C (citrate buffer 10%)	ThermoFisher
Anti-IL-1RN	IL-1R	Polyclonal	1:100	Autoclave 121 °C (citrate buffer 10%)	Sigma
IL-2R.1	IL-2R	Monoclonal	1:1000	PTLink 96 °C	ThermoFisher
8H12	IL-6	Monoclonal	1:40	Autoclave 121 °C (citrate buffer 10%)	ThermoFisher
OTI1D10	IL-10R	Monoclonal	1:250	PTLink 96 °C	ThermoFisher
Ber-H2	TNFR	Monoclonal	Ready to use	PTLink 96 °C	Dako
IFNGR1	IFNγR	Polyclonal	1:200	Autoclave 121 °C (citrate buffer 10%)	ThermoFisher

**Table 2 biomolecules-11-00204-t002:** Taqman probes used for RT-qPCR analysis.

Gene	Full Name	Reference	Source
Gus-β	β-glucuronidase (reference gene)	Oa04828868_m1	ThermoFisher
HPRT-1	Hypoxanthine Phosphoribosyltransferase 1 (reference gene)	Oa04825272_gH	ThermoFisher
IL-1α	Interleukin 1 alpha	Oa04658681_m1	ThermoFisher
IL-6	Interleukin 6	Oa04656315_m1	ThermoFisher
IL-10Ra	Interleukin 10 receptor alpha	Oa04822455_m1	ThermoFisher
IL-10Rb	Interleukin 10 receptor beta	Oa04894070_m1	ThermoFisher
IFNγ	Interferon gamma	Oa04657364_m1	ThermoFisher

**Table 3 biomolecules-11-00204-t003:** Summary of significant (*) or trend (#) statistical results provided in this study.

Group	Marker Assessed (Methodology)	DEX Effect	Brain Area	Statistical Significance
Treated control	IL-1α (IHC)	Increase	MO	* *p* < 0.05
Treated clinical	IL-1R (IHC)	Decrease	Cb	# *p* = 0.082
Treated control	IL-6 (IHC)	Increase	Fc	* *p* < 0.05
		Decrease	O	# *p* = 0.071
Treated clinical		Decrease	O	* *p* < 0.05
Treated clinical	IL-10R (IHC)	Decrease	Cb	*# p* = 0.074
Treated clinical	IFNγ (RT-qPCR)	Increase	Fc	* *p* < 0.05

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
