# Peer review of "Neuroimmune Response Mediated by Cytokines in Natural Scrapie after Chronic Dexamethasone Treatment"

_biomolecules, 2021, doi:10.3390/biom11020204_

Round 1

Reviewer 1 Report

The authors answered to my different remarks and modified the text accordingly.

Reviewer 2 Report

In this re-submitted article, the authors have properly addressed my concerns. My previous concerns have been adequately addressed.

This manuscript is a resubmission of an earlier submission. The following is a list of the peer review reports and author responses from that submission.

Round 1

Reviewer 1 Report

In the manuscript entitled “Neuroimmune response mediated by cytokines in natural scrapie after chronic dexamethasone treatment”, the authors studied the expression of different cytokines and corresponding receptors in a natural model of scrapie-infected sheep, and analyzed the effect of chronic dexamethasone treatment upon these cytokine profiles. The study is mainly based on IHC and RT-qPCR analyses, and the authors detailed and argument in their well–documented discussion the potential effect of DEX on each cytokine their studied.

This study has been performed on animals that have already been described in a previous paper, which includes all the experimental details and analyses of parameters related to their prion diseases (symptoms, disease duration, PrPres accumulation, histopathology…). Subsequently, the authors did provide in the present manuscript absolutely no detail upon these animals and the treatments they received, that is very disconcerting at first reading, notably when the authors claim line 299 “Here, we show that the low dose of DEX chronically administered…”. It is compulsory to mention more obviously that all experimental details have been provided elsewhere, and moreover some information needs to be provided at different levels of the text to be less frustrating (animal genotypes, duration and dose of treatments…). At the level of results, a catalog of observation is detailed: a summary recapitulative table or scheme would be appreciated.

The authors studied 6 cytokines/corresponding receptors: IL-1a, IL-2, IL-6, IL-10, TNF-a and IFN-g. IHC staining was performed for all their receptors, but not for IL-6; why was this receptor neglected?

With IHC techniques, control groups are the ones that provided the main significant differences, and the discussion largely focused on these animals. Conversely, RT-qPCR has only been performed on infected groups, and a significant overexpression of IFN-g was observed: an RT-qPCR analysis of control animals would be of interest.

Concerning statistical analysis, the authors focused on comparing mock- versus DEX-treated groups, either infected or non-infected animals. The authors underlined some significant differences, and also some non-significant trends. According to the graphs, one could wonder about the significance (or trend?) of others, like IFN-gR in MO or IL-1R in FC of control groups. In parallel, why did the authors avoid to compare control versus infected animals, for which graphs seem to indicate significant differences?

Last, the authors highlighted lines 276-278 the higher relevance of their “natural” model in comparison to experimental model. This assertion need to be highlighted by general considerations of comparative immunology between sheep and rodent models, that may provide different cytokine profiles in CNS independently of prion diseases.

Minor comments:

Line 38: Creutzfeldt-Jakob and not –Jacob

Line 141: IHC and not IHQ

Reviewer 2 Report

In the manuscript, authors administered DEX to scrapie-affected or healthy sheep and evaluated their cytokine profiles mainly by immunohistochemistry of the brains. Based on the results, they argued that communications between microglia and astrocyte in the brains of the scrapie-infected sheep were impaired. However, there seems to be no convincing levels of differences in most of the cytokines tested between the groups. The main weakness of their manuscript is the lack of illustration of how reliable their semi-quantitative evaluation method of the intensity of immunostaining. For instance, in Fig 5 that compares expression levels of IL-6 in the occipital lobe between the groups, the graph in Fig5B says that the Untreated Clinical is the highest and the DEX-treated Control is substantially lower; however, in the images of Fig 5A, the DEX-treated Control and the Untreated Clinical apparently have comparable levels of staining, partly due to the difference in the background staining between the images. That undermines the reliability of their evaluation method. The reliability of the evaluation method is particularly critical for their manuscript because even argued significant differences between the groups are often subtle (barely difference of a single score).

Given those technical uncertainties, it is hard to say that the manuscript presents enough evidences for the authors’ arguments at this point.

The Discussion part is full of speculations mostly based on the knowledge from the literature rather than on their own results.

The following are the specific points:

  • In their previous report, administration of DEX showed only a little effect on the disease progression (Ref. 26). In this case, it is conceivable that the cytokine profile at the terminal stage may not be substantially affected by the DEX treatment, because the levels of cytokines would reach the maximal levels, i.e., plateau, by the terminal stage of the illness even if DEX modifies their expression in the earlier phases. Therefore, authors might as well sequentially collect the cerebrospinal fluid from the infected sheep to clarify the relationship between cytokine expression and pathology.

  • In this manuscript, the authors did not statistically analyze the expression of cytokines between untreated control and untreated clinical. To clarify the relationship between cytokine expression and pathology, they should statistically compare cytokine profiles between these two groups.

  • Regarding Fig 1, the authors show the images of immunohistochemical staining of only two groups. They should show the pictures of all groups.
  • Line 325. Authors state that DEX suppressed the expression of IL-1R that might be implicated in the self-renewal of microglia and that the effect might consequently impair the communication between microglia and astrocyte. However, they only observed subtle (statistically non-significant) reductions in the entire expression levels of IL-1R by immunohistochemistry irrespective of cell types, without separately analyzing the expression levels of IL-1R of astrocyte or microglia. The possibility of the impairment of communication between glial cells is a speculation based on works in the literature. However, as the authors featured “an impairment of communications between microglial and astroglial populations" in the abstract as a finding suggested by their works, they should elaborate more on the issue.

  • Line 325. Incidentally, although authors also stated that DEX suppressed the self-renewal of microglia, DEX did not affect the number of microglia in their previous report.

  • Line 366. Although the authors argue that they strongly support the idea that IL-6 is relevant to prion pathogenesis, the presented data do not apparently support the view, because there seems to be no significant difference in the expression levels of IL-6 between untreated control and untreated clinical.

  • Line 388-392. Here, the authors stated that “this is the first study based on a natural model assessing in situ neuroinflammatory activity in prion diseases in relation to chronic anti-inflammatory therapy” and again mention the impairment of communication between glial cells. However, as there was no significant difference in the TNF-R levels among groups after all and they did not measure TNF levels, “This fact encourages another proposal of an impairment of communication between microglia and astroglia” seems to be an overstatement.